# The Potential of OMICs Technologies for the Treatment of Immune-Mediated Inflammatory Diseases

**DOI:** 10.3390/ijms22147506

**Published:** 2021-07-13

**Authors:** Charles Gwellem Anchang, Cong Xu, Maria Gabriella Raimondo, Raja Atreya, Andreas Maier, Georg Schett, Vasily Zaburdaev, Simon Rauber, Andreas Ramming

**Affiliations:** 1Department of Internal Medicine 3—Rheumatology and Immunology, Friedrich-Alexander-Universität Erlangen-Nürnberg (FAU) and Universitätsklinikum, 91054 Erlangen, Germany; CharlesGwellem.Anchang@uk-erlangen.de (C.G.A.); cong.xu@uk-erlangen.de (C.X.); mariagabriella.raimondo@uk-erlangen.de (M.G.R.); georg.schett@uk-erlangen.de (G.S.); simon.rauber@uk-erlangen.de (S.R.); 2Department of Internal Medicine 1, Friedrich-Alexander-Universität Erlangen-Nürnberg (FAU) and Universitätsklinikum, 91054 Erlangen, Germany; raja.atreya@uk-erlangen.de; 3Computer Science, Friedrich-Alexander-Universität Erlangen-Nürnberg (FAU), 91054 Erlangen, Germany; andreas.maier@fau.de; 4Max-Planck-Zentrum für Physik und Medizin, 91054 Erlangen, Germany; vasily.zaburdaev@fau.de; 5Department of Biology, Mathematics in Life Sciences, Friedrich-Alexander-Universität Erlangen-Nürnberg (FAU), 91054 Erlangen, Germany

**Keywords:** immune-mediated inflammatory diseases (IMIDs), single-cell RNA sequencing, spatial sequencing, advanced imaging technologies

## Abstract

Immune-mediated inflammatory diseases (IMIDs), such as inflammatory bowel diseases and inflammatory arthritis (e.g., rheumatoid arthritis, psoriatic arthritis), are marked by increasing worldwide incidence rates. Apart from irreversible damage of the affected tissue, the systemic nature of these diseases heightens the incidence of cardiovascular insults and colitis-associated neoplasia. Only 40–60% of patients respond to currently used standard-of-care immunotherapies. In addition to this limited long-term effectiveness, all current therapies have to be given on a lifelong basis as they are unable to specifically reprogram the inflammatory process and thus achieve a true cure of the disease. On the other hand, the development of various OMICs technologies is considered as “the great hope” for improving the treatment of IMIDs. This review sheds light on the progressive development and the numerous approaches from basic science that gradually lead to the transfer from “bench to bedside” and the implementation into general patient care procedures.

## 1. Introduction

Immune-mediated inflammatory diseases (IMIDs) comprise a large number of chronic inflammatory conditions such as inflammatory bowel diseases (IBD; Crohn’s disease, ulcerative colitis), chronic inflammatory arthritis (rheumatoid arthritis, spondyloarthropathies), and multiple sclerosis. The clinical course of these debilitating diseases is typically marked by relapsing exacerbations, causing lifelong morbidity, disability, and heightened risk of disease-associated mortality [1]. Optimized anti-inflammatory therapy is therefore essential in the management of these disorders, mainly driven by a dysregulated immune response that initiates and perpetuates the inflammatory reaction. Growing insights into underlying immunopathogenic mechanisms have led to the advent of targeted immunotherapies, which selectively inhibit crucial mediators of the inflammatory process. Nevertheless, only a subgroup of treated patients respond to initiated therapies [2,3], reflecting the still incompletely understood immunopathogenesis of these diseases. Ongoing research findings have indicated that distinct immune cell populations and immune-regulatory pathways play a crucial role in the acute and chronic inflammatory reactions [4]. Herein, technological advancements such as high-throughput omics data generation have been used as powerful tools to discover genomic mutations, epigenomic modifications, abnormal transcription/translation, and cellular tissue distribution. Next-generation sequencing (NGS), proteomics, and metabolomics have catalyzed research and revealed disease-related associations and patterns that advance our understanding of the inflammatory process. Furthermore, the quality of novel imaging approaches to visualize the inflammatory process has dramatically increased. Together, we have powerful tools in our hands to start the era of personalized medicine. However, the complexity and amount of the generated datasets necessitate the need to aid the biologist and physician in the interpretation of the datasets and images, as well as hampering the direct translation into clinical applications, which would be important for the fast establishment of personalized medicine. In the case of IMIDs, treatment decision making is still random, without clear rational arguments, despite the exponential increase of use of NGS in basic science. RNA sequencing, proteomics, metabolomics, whole genomic sequencing, and whole exome sequencing are well-established techniques that provide a large number of results and are actually the most used omics techniques, but are also continuously improving and evolving. In this review, we address recent advances in high-throughput omics technologies and reflect the slow but increasing implementation in clinical practice.

## 2. On the Path of Single-Cell Omics

The beginning of the 21st century marked a new chapter for genomic research; next-generation sequencing (NGS) technology made it possible to routinely generate billions of reads and to quantify millions of transcripts in parallel. It first started with the analysis of cell mixtures, but further technological developments; protocol optimizations; and, above all, the increasing commercialization with upcoming simplifications and standardizations of the experimental procedures enabled the generation of datasets on a single-cell level [5]. This development of new single-cell technologies has uncovered a new view on the basic unit of life, a thus far undetected high degree of heterogeneity that exists between individual cells. The use of single-cell technologies could show that, e.g., the heterogeneity of T cells goes far beyond the division into T helper (Th)1/Th2/Th17 and regulatory T cells). Recently, mucosal-associated invariant T (MAIT) cells and their functional roles in tissue repair as well as several pathologies, e.g., colitis, arthritis, and multiple sclerosis, have been tackled [6]. Innate lymphoid cells (ILCs) were genotypically separated from natural killer (NK) and T cells [7,8,9,10]. Functional studies revealed different types of ILCs with the ability to resolve inflammation [11], but also to amplify pathogenic processes in IMIDs [12,13]. Furthermore, fibroblasts are not considered any longer as bystander cells, but have been divided into several subpopulations according to their functional implications. Wohlfahrt et al., showed that extracellular matrix production is under control of a transcriptional network including PU.1 [14]. The discovery of PU.1 as a potential profibrotic transcription factor was based on a bioinformatic analysis of bulkRNAseq datasets from 97 patient samples and opened the door to the development of a new class of PU.1-inhibiting substances for the treatment of fibroinflammatory IMIDs. ScRNAseq analyses revealed NOTCH3 and THY1 as markers of distinct fibroblast subtypes in the synovial membrane, which are implicated in persistent inflammation and joint destruction [15,16]. The discovery of this subpopulation opens up completely new possibilities for the treatment of inflammatory arthritis with fibroblasts as main target. Genomics, transcriptomics, epigenomics, etc. provide researchers with opportunities to interrogate the heterogeneity of single cells at unprecedented depth. Nonetheless, the abundance of mRNA transcripts is not strictly reflective of the level of present functional protein. Post-transcriptional and post-translational modifications further tune biological processes and cellular phenotypes. The combination of single-cell transcriptomic and proteomic technologies will be the key to a comprehensive picture of the heterogeneous regulatory mechanisms of an individual cell in the future. Humby et al., recently showed that RNA sequencing-based stratification of patients with rheumatoid arthritis can guide physicians to choose the most effective therapy [17]. On the protein level, there are numerous high-dimensional procedures such as highly multiplexed imaging, mass cytometry, and numerous other single-cell proteomic platforms in the pipeline. Increasing technical availability, decreasing costs, and advanced automation generate tremendous new possibilities for personalized medicine (Figure 1).

## 3. Single-Cell RNA Sequencing (scRNA-Seq)

ScRNA-seq was first established in 2009 [18]. Various techniques have been developed: Some use barcoding of RNAs with unique molecular identifiers (UMI), allowing for the distinguishing between original mRNA reverse transcripts and library preparation-based PCR amplicons. Other approaches rely on full-length sequencing of cDNA fragments, allowing for a higher sensitivity or detection of unknown mRNA sequences [19,20,21]. Furthermore, technologies can be distinguished as to whether cDNA synthesis occurs in a water-in-oil emulsion, encapsulating single cells with master mix and barcoded solid support (droplet), or if cells are sorted into individual wells of multi-well plates or microfluidic chips. Technologies relying on either of the approaches have been commercialized, facilitating scRNA-seq in becoming a mainstream technology. For droplet-based approaches, specific instrumentation is required, which might cost up to USD 75,000 and/or proprietary kits allowing for the generation of single-cell libraries for USD 0.15–0.20 per cell. Droplet-based approaches are considered less sensitive but allow high-throughput. In conventional sorting-based approaches, costs per single cell library might increase to up to USD 20, but usually no further instrumentation is needed, as mostly in droplet-based approaches also, cells of interest will be sort purified. Initially, fluorescent dyes have been used for barcoding droplets. However, their use is limited for encoding only a few hundred samples, owing to the dynamic range of the optical setup. Larger sample numbers can be achieved with DNA oligonucleotides instead of fluorescent dyes whose barcoding capacity corresponds to 4n (with n being the number of nucleotides) and is hence almost unlimited.

InDrop, Drop-seq, and 10× Genomics Chromium are the three most frequently used systems until now. Besides differences between the systems with regard to bead manufacturing and barcode design, the cDNA conversion distinguishes in particular the systems [22]. Whereas Drop-Seq only captures the transcripts without cDNA conversion, reverse transcription is carried within droplets for 10× and inDrop before demulsification. Cells with a high enzymatic activity such as neutrophils interfere with the RT and result in low cDNA yield. In such situations, Drop-seq is the better variant because the sensitive step of cDNA conversion does not take place in the droplet but starts later after demulsification of the cells and therefore in the absence of cell specific enzymes.

Besides understanding the composition of the inflammatory niche, single-cell transcriptomic analysis can guide successful therapeutic interventions for patients with IMIDs. Kim et al., recently described the successful use of tofacitinib in a patient with drug-induced hypersensitivity syndrome after performing an scRNAseq analysis of the patient’s skin and blood [23]. ScRNAseq has also been successfully used in other cases such as metastatic ALK-positive lung cancer [24]. Costs of sequencing applications have decreased significantly over the last 20 years. According to the National Human Genome Research Institute (NHGRI), the price of 1 raw mega-base (approximately 3400 reads) was USD 10,000 in 2001 but dropped to USD 0.01 in 2020 [25]. Initially, the costs of sequencing developed according to the popular “Moore’s law”, which describes cost reduction by half every second year [26]. This cost degression changed in 2008 by a technology switch from Sanger sequencing to NGS, inducing a knockdown of costs and subsequently allowing for new applications. The major technical hurdle of large-scale scRNA-seq studies is the cost of preparing and sufficiently sequencing large numbers of single-cell libraries, hence limiting its transition into clinical routine diagnostics. ScRNAseq in diagnostic application is likely to be reserved for rare diseases for which controlled cohort studies can only be carried out to a very limited extent. Many IMIDs belong to the rare diseases, and therefore this new diagnostic tool can enable further, significant therapeutic successes in the future.

## 4. Spatial Sequencing

Despite an increasing number of studies employing scRNA-seq in IMIDs, spatial information of the tissue context is missing. However, spatial gene expression heterogeneity does not only play an essential role during organ development but also in various pathological processes of IMIDs. Macrophages are able to protect tissue niches but also cause the migration of further immune cells to open up the niche for inflammatory responses [27]. Fibroblasts build lining layers surrounding the joint cavitation. Local destruction of this physiological barrier primes inflammatory processes [28]. Therefore, tissue-specific insights in IMIDs are of interest in order to select the one with the highest chance of success from the available therapeutics. Spatial sequencing emerged to address this problem. Early attempts were based on multiplexed single-molecule fluorescent in situ hybridization (smFISH) via spectral barcoding. Over the past years, smFISH evolved rapidly from detecting few genes to the whole transcriptome level (smFISH -> seqFISH, MERFISH -> SeqFISH+) [29]. Despite these advances, SeqFISH+ could not yet gain broad acceptance due to its high technical requirements including high-sensitivity single-molecule fluorescence imaging systems. The technique enables the detection of up to 10,000 different transcripts on a (sub)cellular level in situ but does not allow for a complete or even de novo sequencing such as NGS. The transcripts are detected by a pre-defined set of probes, and thus FISH-based spatial gene expression analysis rather resembles to the micro array technique. Each transcript is hybridized with about 24 primary probes of 28 nt gene-specific hybridization regions binding to exonic regions. Primary probes contain on 5′ and 3′ binding sites for secondary read-out probes. Subsequently, rounds of hybridization with shorter read-out probes (15 nt) coupled to fluorescent dyes, probe stripping, and imaging steps are used to generate a virtual stack containing 60 pseudo-color channels. From these images, similar to Sanger sequencing, the color sequence for each pixel reveals the transcript. The major advantage of this approach is that it is capable of resolving transcripts at subcellular resolution. Other approaches rely on NGS for transcript detection and need hybridization of spatially encoded barcodes. In the 10× Genomics Visium assay, those barcodes are spotted on microscopic glass slides on 6.5 × 6.5 mm areas in which tissue sections are placed and hybridization of mRNA takes place. Each area contains 5000 printed barcoded mRNA capture probes in 100 mm center-to-center distance from one to the other. The system, therefore, achieves a maximum resolution of 50 μm, which is far away from single-cell level and limited in terms of use for interactome analyses. Instead of printing regional barcoded RT primers onto a glass slide, densely barcoded bead arrays have been dispensed on a glass surface in Slide-seqV2 [30]. This technique reaches a resolution of 10 μm, meaning a near to single-cell level and an almost 100% higher rate of RNA capture efficiency compared to 10× Visium. Seq-Scope is another tool for high resolution spatial transcriptomics, enabling the visualization of transcriptomic heterogeneity at the cellular/subcellular level. In contrast to Slide-seq V2, which is based on barcoded beads, seq-scope depends on a solid-phase amplification of randomly barcoded single-molecule oligonucleotides spotted on a Illumina flow cell [31]. Deterministic barcoding in tissue for spatial omics sequencing (DBiT-seq) is another method that uses a microfluidic chip with parallel channels on top of a tissue section [32]. For each channel, oligo-dT-labeled barcodes are streamed across the tissue. After removing the first chip, a second microfluidic chip is rotated 90° to the first and placed on the tissue. Again, barcodes are streamed through the parallel channels, resulting in a mosaic of 10 µm side rectangles of barcodes from the first and second streams. The barcodes anneal to mRNAs to initiate in situ RT, resulting in stripes of barcoded cDNAs inside the tissue. A distance of max. 10 μm between the channels has been established, resulting in a resolution of 20 μm, which is close to the single-cell level. With regard to IMIDs, Carlberg et al., explored inflammatory signatures in arthritic joint biopsies with spatial transcriptomics [33]. Direct diagnostic applications are still missing. However, joint biopsies are frequently very limited of size and therefore often do not allow the use of scRNAseq. Spatial sequencing could represent a new opportunity here to obtain transcriptional data from small tissue samples, particularly when making therapy decisions for IMID patients.

## 5. On the Road of Single-Cell Proteomics

On the protein level, mass cytometry has evolved as a valuable high-throughput assay for single cells and is recently also available for the analysis of tissue sections (imaging mass cytometry (IMC)). Mass cytometry/IMC is a technique that resembles flow cytometry but uses rarely abundant metal-coupled antibodies to stain for proteins instead of fluorescent dyes. Since these metals are usually not present in biological specimens, mass cytometry usually has a minimal background and a good signal-to-noise ratio. Technically, single cells are ionized with inductively coupled argon plasma, and then the ions are separated by mass (TOF mass spectrometry) and quantified. Mass cytometry allows for the generation of high-throughput data in a very short period of time. In IMC, FFPE or frozen tissue sections are stained with metal-labeled antibodies. The resolution is about 0.5–1 µm, allowing for subcellular localization of protein expression. Currently, panels including about 40 different antibodies are feasible for IMC and mass cytometry. IMC is a powerful tool to detect proteins in a directed manner but is limited by the availability and applicability of the metal isotopes. Furthermore, other macromolecules such as lipids or small metabolites cannot be detected. Unbiased approaches are based on mass spectrometry, e.g., matrix-assisted laser desorption ionization (MALDI) imaging, which takes advantage by measuring metabolites but does not reach single-cell resolution [34]. Yet, undirected deconvolution of MALDI spectra to single molecules can be challenging. Single-cell proteomics by mass spectrometry (SCoPE2) is another technique in which cells are sorted into multi-well plates before performing tandem mass spectrometry combined with liquid chromatography (LC–MS/MS) analysis, virtually allowing for single cell proteomics [35]. Lastly, proteo-genomics uses DNA barcode-tagged antibodies to detect protein expression by sequencing; e.g., GeoMx Digital Spatial Profiler technology from NanoString [36] can easily be combined with RNA in situ hybridization (ISH). As the number of possible DNA barcodes is virtually unlimited, this technique is mostly limited by the gradually increasing costs for antibodies/oligonucleotides and sequencing and the relatively low amount of cells compared to mass cytometry. Initial successes in the clinical use of mass cytometry as a diagnostic tool were achieved in the treatment of malignant melanoma. Krieg et al., analyzed 40 cryopreserved PBMC samples from the blood of a cohort of 20 patients with melanoma before and after initiation of anti PD-1 immunotherapy [37]. They identified CD14+CD16-HLA-DRhigh monocytes as strong predictors of progression-free and overall survival in response to anti-PD-1 immunotherapy. In psoriatic arthritis, Penkava et al., used IMC for measuring clonal expansions of pro-inflammatory synovial CD8 T cells in synovial fluid [38].

## 6. New-Generation Microscopy

Non-invasive imaging modalities such as CT, PET, and MRI have been extensively used in clinical practice and greatly facilitate diagnosis and evaluation of treatment of autoimmune diseases. However, their imaging lateral resolution is within the range of 50–100 μm [39]. Therefore, cellular and subcellular changes during the onset and development of immune diseases are hardly resolved by these techniques. Understanding these changes will help to design novel therapeutic strategies against autoimmune diseases. Fluorescence microscopy is a widely used scientific imaging method to image single cells and subcellular structures with a spatial resolution of tens to hundreds of nanometers. As with other imaging modalities, many different fluorescence microscopy methods have been developed for biological imaging. As discussed in this section, confocal microscopy, two-photon microscopy, light sheet microscopy, and super-resolution microscopy have been used widely in autoimmune disease related studies.

### 6.1. Confocal Laser Scanning Microscopy (CLSM)

Confocal microscopes focus a laser beam on a small focal spot that scans through the sample and detects fluorescent light emitted by the sample. The use of a detection pinhole permits light only from the focal plane to reach the detector, and therefore enables optical sectioning. Consecutive imaging at different depths allows for three-dimensional (3D) volumetric imaging of samples [40]. Under optimal conditions, confocal microscopy could reach 180 nm high spatial resolution for imaging subcellular or molecular structures [41,42]. With additional multiplex imaging abilities, confocal microscopy has become a very popular method for biological imaging. Furthermore, in vivo confocal microscopy has become an essential tool to decipher temporal and spatial information simultaneously for cells of interest [43,44,45]. In the research field of autoimmune diseases, confocal microscopy revealed a number of fundamental understandings of how cellular alterations of immune cells are linked to the onset or development of diseases. In type 1 diabetes mice, for example, in vivo confocal microscopy revealed that T cells infiltrate first into the islets and initiate the autoimmune process [46]. Together, CLSM is a powerful method for 3D imaging of immune cells at the cellular level. Its imaging depth, however, is limited to tens of microns due to tissue scattering [47]. Therefore, it is not an ideal method for 3D imaging of entire thick samples, as sectioning fixed tissue is often required for imaging structures in deep layers. Furthermore, a confocal microscope illuminates through the specimen with a double inverted cone of light. Fluorescence dyes within the entire laser illuminated volume are easily photobleached due to overexposure to the laser light, which further limits 3D imaging. In the case of long-term live-cell imaging, long exposure to laser light also causes phototoxicity to imaged cells.

### 6.2. Multi-Photon Laser Scanning Microscopy (MPLSM)

Some of these imaging disadvantages can be mitigated by multiphoton laser scanning microscopy (MPLSM), in which a fluorophore absorbs two or three photons emitted by a pulsed infrared laser source. The infrared excitation light penetrates tissue deeper than visible light at shorter wavelengths used in confocal microscopy due to less tissue scattering. In addition, MPLSM restricts the fluorophore excitation in a very small focus and thus results in optical sectioning without the need of a pinhole used in confocal microscopy to block the light out of focus. Together, MPLSM enables the deep tissue penetration depth, the lack of out-of-focus bleaching, and the reduced photo-damage while maintaining very good spatial and temporal resolution. Two-photon laser scanning microscopy (TPLSM), the most popular form of MPLSM, is therefore widely used for 3D imaging and intravital imaging of immune cells in vivo [48]. In experimental autoimmune encephalomyelitis (EAE), a popular model for multiple sclerosis [49], the detailed processes of T cell infiltration into the central nervous system (CNS) and their activation were revealed by intravital imaging via TPLSM [50,51]. Furthermore, TPLSM imaging facilitated direct evaluation of therapeutic effects of anti-integrin α4 antibody and calcium inhibitor BZ194 in EAE model [51]. Therefore, intravital imaging allows for the direct observation of cell dynamics in living organs and serves as a powerful tool to evaluate potential therapeutic treatments for autoimmune disease. As with CLSM, the imaging depth of TPLSM is largely compromised by the signal-to-background ratio (SBR) of the excitation due to tissue scattering. Three-photon microscopy laser microscopy (3P-LSM) invented recently utilizes longer wavelengths (>1300 nm) that decrease excitation light scattering as well as dramatically reduces the out-of-focus background, improving the SBR by orders of magnitude when compared to TPLSM [52,53]. 3P-LSM allows for imaging single cells at 2 mm deep inside the tissue.

The generation of a nonlinear optical process, named harmonic generation (HG), is another advantage of MPLSM imaging. Photons simultaneously interact with biological materials and convert into one photon in HG. Secondary harmonic generation (SHG; two photons converted into one photon) and third harmonic generation (THG; three photons converted into one photon) are the most common forms of HG used for label-free imaging by MPLSM, as signals come from the optical property and the structure of the material. For biomedical applications, HG is very useful in the imaging of connective tissue, collagen fibers, muscle, inflammatory cells, and blood vessels in vivo and in vitro [54]. For instance, adipocytes, collagen, nerve fibers, blood vessels, and muscle can be visualized simultaneously without labeling in the native dermis [55]. The extracellular matrix (ECM) is altered by cytokines and proteases produced by infiltrated immune cells in many autoimmune diseases [56]. Therefore, deciphering the mutual interaction of immune cells and ECM during the onset and development of autoimmune diseases by MPLSM may provide novel insights to modulate disease progression. Together, deep tissue imaging and HG imaging conferred by MPLSM will still be versatile techniques for autoimmune disease-related studies in the future.

### 6.3. Light-Sheet Fluorescence Microscopy (LSFM)

While CLSM and MPLSM enable imaging with great spatial resolution and penetration depth, the acquisition speed of both microscopies is limited because the image is built up one voxel at a time. In order for this limitation to be addressed, light-sheet fluorescence microscopy (LSFM) was developed. The concept of LSFM was introduced more than 100 years ago [57] and has been advanced in recent years for biological imaging [58]. The processes of sample illumination and fluorescence detection are decoupled in LSFM, in which a sample is illuminated with a thin sheet of laser light and fluorescent light emitted from within the illuminated plane is imaged with a wide-field camera oriented orthogonally to the light sheet. Light sheets are usually generated by focusing a laser with a cylindrical lens. Using a camera confers LSFM capability of at least 100x faster high-speed imaging than point-scanning methods offered by CLSM and MPLSM because all pixels from a single plane are now imaged simultaneously [59]. Additionally, LSFM enables optical sectioning via a light sheet confined to the plane of interest for fluorescence excitation while precluding the illumination of out-of-focus structures. By moving samples up or down through a fixed light sheet and taking a digital image at each height enables 3D imaging. Combined with advanced tissue clearing methods (reviewed elsewhere [60]), LSFM has been used successfully to image individual organs [61], a whole mouse [62], and intact human organs such as kidney and lung [63,64]. In RA, infiltrating macrophages mediate inflammation and contribute to the disease progression. Recent work using LSFM discovered CX3CR1+ tissue resident macrophages, however, providing an anti-inflammatory barrier. These synovial macrophages are located in dynamic membrane-like structures, which isolate the joint from the surround tissues [27]. A similar anti-inflammatory function was also revealed in the dermis resident macrophages [65].

### 6.4. Super-Resolution Microscopy (SRM)

An inherent drawback of fluorescence microscopy is its spatial resolution, which is limited by the diffraction of light. The diffraction limit is about 200–300 nm in the lateral direction and 500–700 nm in the axial direction. Many subcellular structures will be too small to be resolved under this resolution. The diffraction limit was bypassed by the invention of several SRM technologies, such as stimulated emission depletion (STED) microscopy [66,67], structured illumination microscopy (SIM) [68], reversible saturable optically linear fluorescence transitions (RESOLFTs) [69], saturated structured illumination microscopy (SSIM) [70], stochastic optical reconstruction microscopy (STORM) [71], photoactivated localization microscopy (PALM) [72], and fluorescence photoactivation localization microscopy (FPALM) [73]. These technologies have been used to study the subcellular organization and function of molecules, e.g., T cell receptor forming nanoclusters together with Linker for activation of T cells [74,75]. Understanding how the sensitivity of T cell responses is mediated by these molecular organizations will help to develop the strategies for rendering the immune system less sensitive in the case of autoimmune diseases. Further developments such as interferometric scattering microscopy (iSCAT) allow for the visualization of small vesicles and proteins without the need for a fluorescent label [76]. Brillouin microscopy will take mechanical properties of the tissue into account for cellular function and disease on a microscopic level [77].

### 6.5. Limitations in the Direct Translation of Omics Data into Clinical Use

The combination of molecular omics data profiling and improved imaging techniques provides unique advantages for clinical diagnosis, monitoring, and therapy of IMIDs, which could lay the basis for the optimized care of patients with IMIDs. However, these high-end technologies are of limited utility without complementary methods for interpretation of the collected data to decipher the inherent complexity of the investigated diseases. To extract detailed information and draw conclusions with wider applicability to biological processes, datasets have to be integrated and analyzed as a holistic system. For example, the Standardization of Clinical Testing (Nex-StoCT) and the U.S. Centers for Disease Control and Prevention (CDC) have collaborated to establish measures to ensure that analytical approaches of NGS data are valid, and that tests based on NGS meet existing standardized clinical laboratory requirements [78]. The consortium of the Genetic European Variation in Disease (GEUVADIS) demonstrated that it was practically possible to reproduce RNA sequencing experiments across laboratories by sequencing lymphoblastoid cell lines from 465 patients in seven sequencing centers using the same platform [79]. From this study, the consortium proposed a set of quality checks to address technical biases in RNA sequencing data, such as GC content, fragment size, transcript length, and percentage of reads mapped to exons annotated hitherto [79].

Bioinformatics challenges also explain why omics technologies are yet to gain broad clinical applicability. Taking the context of RNA sequencing, one major challenge includes the fact that there is currently no harmonized body to lay down standards that will ensure the analytical quality of RNA sequencing analysis pipelines. Another challenge has to do with the abundance of software tools and their combinations for the analysis of RNA sequencing data, and in third place, highly complex pipelines consisting of the chaining together of tools that are largely independently developed, maintained, updated, and licensed [80]. Very few of the algorithms developed to analyze NGS data are capable of producing reliable clinical predictions [81].

## 7. Systems Biology Approach to Mine through Complex Omics Data

The advances in gathering omics data that are scaled down to a single cell led to enormous, to some extent even boundless amounts of information that is impossible to comprehend without developing dedicated tools and methods. The desire to use this rich information and uncover new, yet unknown dependencies, regulatory mechanisms, and patterns greatly influenced the development of the new research field of Systems Biology. Not pretending for the generality of the definition [82], Systems Biology is a holistic approach that integrates broad range of available information about living systems and then uses and develop methods that could simplify their complexity down to comprehensible interdependencies and regulatory pathways, ultimately leading to the emerging understanding of biological functions [83]. Its applications in the field of medicine are gaining recognition as Systems Medicine [84,85,86] and are spearheaded by Cancer Systems Biology [87] and Systems Immunology [88,89].

The initial step of the bottom-up approach of Systems Biology is the collection and import of datasets on transcriptome, proteome, and metabolome (RNA-Seq, ChIP-seq, quantitative mass spectrometry, etc.), previously acquired in the contexts relevant to a concrete problem or medical condition in question, frequently also for multiple species (i.e., human, mice) [90]. Already, this step requires specialized software to import data, as well as for accessing and harmonizing data from various existing data repositories, and finally preparing it for the next conceptual step of generating interactome network. Interactome network is combined of nodes and links representing possible relations between them. Nodes are quite broadly defined as representing molecules (proteins, mRNA, miRNA, transcription factors, enzymes, metabolites, etc.), cells and cell types, and biological functions. The links (directed or undirected) between the nodes denote their corresponding interactions. The typical interaction types include protein–protein, DNA–protein or gene regulatory interactions of transcription factors, co-factors and genes, mRNA–mirRNA, and biochemical interactions as in substrate/product/enzyme reactions. Due to advances in algorithms, software, and spreading standardization of data storing, the network assembly is gradually turning from a largely manual task towards the semi-automated analysis [91,92]. A network size for a typical particular application easily reaches several thousands of nodes, and for a dedicated, whole-organism analysis, for example, budding yeast, the number of links is counted in the hundreds of thousands [93]. The core creative discovery step of the approach occurs during the comprehensive analysis of the network by using advanced mathematical tools of the network theory [94]. Only with its help does it become possible to identify network subclusters as linked to a given condition, search and classify regulatory motifs (i.e., feedback loops, not to be confused with transcription factor binding sequences), and group nodes according to a specific biological function or a phenotype. Whenever a distinct and substantially smaller subset of nodes of the full network is identified according to a certain criterion, it then is redefined as the so-called core network that could, for example, produce the most significant changes under a certain condition, or describe a certain phenotypic change. Having a much more observable number of nodes and links such core networks are amenable to further mathematical analysis. To this end, a system of differential equations that describe the temporal evolution of nodes and their interrelations can be set up and studied as a function of parameters and network topology. A mathematical model, relying on a large body of knowledge of dynamical systems, can then deliver rigorous predictions on possible attractors of the dynamics and their stability, existence of periodic patterns, or switching of the regimes and their response to perturbations [95].

There is a good number of recent studies employing the systems biology approach in the context of immunology and inflammation. A network of signal transduction pathways for inflammatory macrophages, containing 1122 molecule species (nodes) and 2705 reactions (links), was manually constructed and annotated [96]. By considering three lung infection scenarios, researchers identified a regulatory core of 41 factors, including TNF, CCL5, CXCL10, IL-18, and IL-12 p40, as well as 140 drugs targeting 16 of them. A comprehensive study [97] used the systems biology approach to analyze an extensive dataset on six autoimmune diseases of multiple sclerosis, systemic lupus erythematosus, juvenile rheumatoid arthritis, Crohn’s disease, ulcerative colitis, and type 1 diabetes, focusing on intracellular regulatory mechanisms in peripheral blood mononuclear cells (PBMCs). It was shown that chemokines such as CXCL1-3, 5, and 6 and the interleukin IL-8 were differentially expressed in PBMCs of patients with diseases, and more generally, similar cellular processes related to cell proliferation, inflammatory response, and apoptosis tend to be differentially expressed in PBMC. In a very recent effort [98], a fully annotated, expert validated, state-of-the-art knowledge base for RA in the form of a molecular map was developed and is available online at ramap.elixir-luxembourg.org. The corresponding network based on 353 publications contains 506 nodes (303 proteins, 61 complexes, 106 genes and 106 RNAs, 2 ions, and 7 simple molecules), 446 reactions, and 8 phenotypes. This open-access knowledge base allows for easy navigation and for the search of molecular pathways implicated in the disease.

Thus, the principal tools and methods of Systems Biology are in place to be applied also to the problem of inflammatory diseases, while the computational algorithms and database integration continue rapid development. The major challenges still exist, as for example in combining different layers of datasets (i.e., omics and imaging data). Importantly, the experimental validation of network analysis predictions still remains a crucial step within this approach.

## 8. Perspectives to Take Advantages of OMICs Technologies in Daily Clinical Practice

The integrated analysis of protein–protein, gene regulation, and coexpression network models, alongside gene expression data, can lead to the discovery of the most relevant genes and pathways in IMIDS, as well as causal mechanisms and their topological properties [99,100,101,102,103]. On the other hand, enrichment analyses can enable the tracing of pathological mechanisms from the DNA level to the pathophysiological level [99]. For example, assays such as RNA-seq and Chip-seq, which measure gene expression and regulation of gene expression, respectively, have been applied to elucidate tissue-specific signatures of genomic regulation [104]. The clinical application of multi omics data poses major challenges including the joint requirement for expertise and advanced facilities in statistics, biology and computer sciences, as well as the interpretation and therapeutic actionability of molecular findings. However, the integration of multiple omics data types holds a large amount of promise, as it can help to provide a holistic picture of diseases with unprecedented details and guide therapy [99]. Furthermore, DNA and RNA sequencing have been successfully applied to identify gene signatures in tumor versus normal control sample comparisons, which guided therapies that lead to tumor regression [105,106]. The same approaches are likely to yield success if transferred to IMIDs (Figure 2).

Taken together, the synergistic competence in molecular biology, bioinformatics, biostatistics, bioethics, computer sciences, mathematics, and medicine might enable the expansion of our understanding of the underlying disease mechanisms in the clinical fields of IMIDs and could lead to advancements in inflammation research. Artificial intelligence (AI) and machine learning (ML) are additional techniques that have the capacity to identify and uncover clinically relevant patterns and interdependencies amongst the acquired information and take confounding factors into account [107,108]. They differ from traditional statistical methods as they focus on prediction and classification from high-dimensional data, rather than pure inference on typically univariate data. As successful ML requires robust and sufficient data, from which it can learn, well established omics/imaging technologies in combination with comprehensive clinical and outcome information provide an apt basis for application of ML algorithms. A key challenge for most modern AI methods is the requirement of large amounts of training data in general and task-agnostic learning approaches [109]. In medical research, this requirement is often not met. Yet, there is a broad body of medical domain knowledge in form of molecule interactions, biological processes, imaging data, and general high-level correlations that is typically not suited for inclusion in general ML approaches. Large collaborative efforts such as the Human Cell Atlas (HCA) [110] are currently dealing with the question of how information can be integrated across different platforms. They underline the importance of computationally mapping data from different modalities to a reference [111]. In order for us to be able to integrate data from spatial and non-spatial scRNAseq data, new calculation methods for the creation of such atlases will be of fundamental importance. The interdisciplinary setup of biologists, computer scientists, mathematicians, and physicians could ultimately lead to the implementation of precision medicine concepts and thus clinically meaningful benefits for the individual patient with IMIDs (Figure 3).

## Figures and Tables

**Figure 1 ijms-22-07506-f001:**
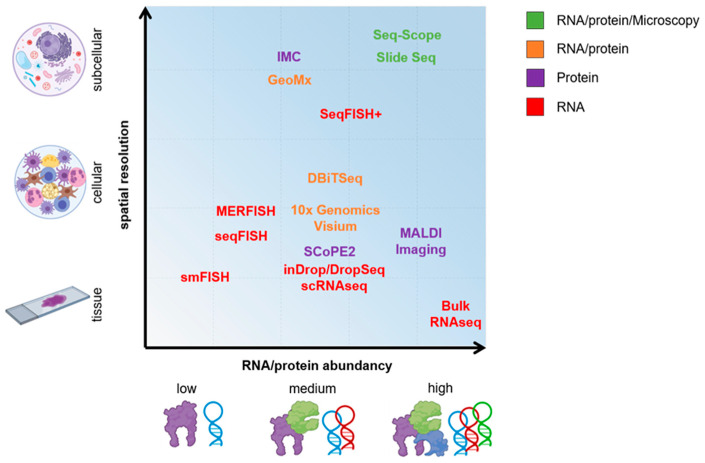
Omics methodologies with regard to RNA/protein abundancy and spatial resolution. Certain technologies allow for a combinatory assessment on RNA and protein level, and microscopic analyses. DBiTSeq, deterministic barcoding in tissue for spatial omics sequencing; FISH, fluorescence in situ hybridization; GeoMx Digital Spatial Profiler technology from NanoString; IMC, imaging mass cytometry; MALDI, matrix-assisted laser desorption/ionization; MERFISH, multiplexed error-robust fluorescence in situ hybridization; Seq, sequencing; smFISH, single-molecule fluorescence in situ hybridization.

**Figure 2 ijms-22-07506-f002:**
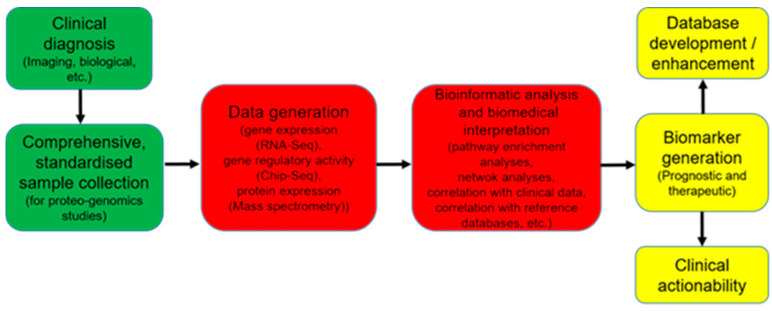
OMICs in clinical application. A schematic view.

**Figure 3 ijms-22-07506-f003:**
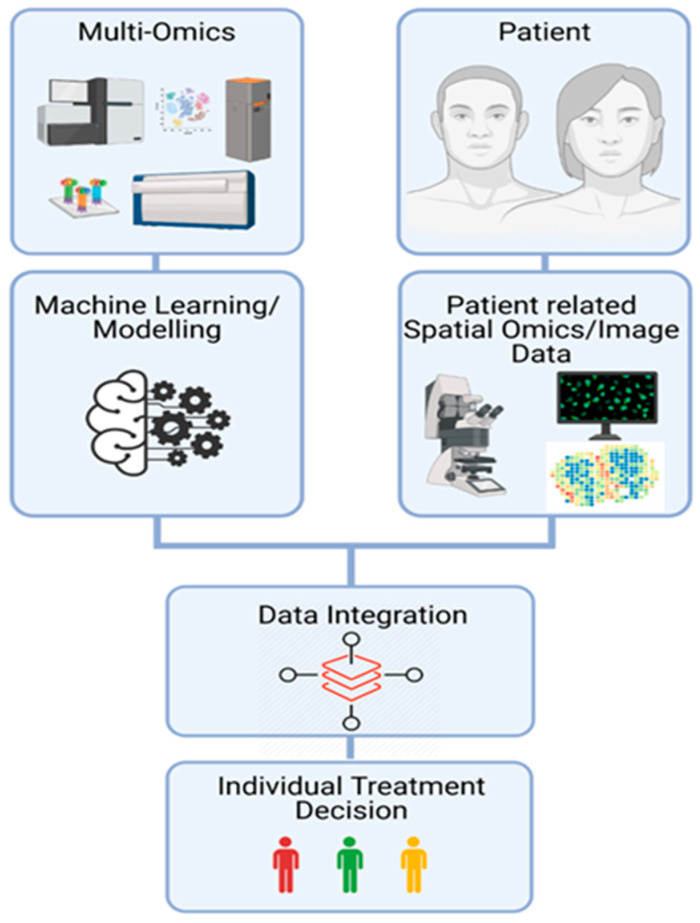
Data Integration. Interdisciplinary setup of multi-omics, spatial transcriptomic, and imaging data including machine learning techniques, providing a platform for individual treatment decisions.

## Data Availability

Not applicable.

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
