# Peer review of "The Potential of OMICs Technologies for the Treatment of Immune-Mediated Inflammatory Diseases"

_ijms, 2021, doi:10.3390/ijms22147506_

Round 1
Reviewer 1 Report
The manuscript entitled “Getting closer to translational bioinformatics- The potential of OMICs technologies and artificial intelligence in the treatment of immune mediated inflammatory diseases” is a manuscript that tries to present, also from a technical point of view, a series of omics techniques and to explain their contribute in the field of immune mediated inflammatory diseases.
The idea behind the work is good because it would be useful to have an overview of the most recent techniques and to understand how they could be effectively useful in the clinical application. However, I think that the manuscript results quite inhomogeneous and lacks a well-organized structure.
There are issues that should be addressed in order to make the work suitable for publication.
First of all, there is not a Conclusion/discussion paragraph: especially in a review, this part must be included because it must present the point of view of the authors on the subject of the review
Another issue that arises is the difference in the matter between the “sequencing” paragraphs (from 2 to 6) and the “microscopy” paragraphs (from 7 to 11). Paragraphs 2 to 6 are just a description of the experimental approach presented: in some cases, the explanation is even too much detailed (for example from lane 219 to lane 227) and there are just a few examples of the various application of the techniques for the study of IMIDs. On the other hand, I think the paragraphs from 7 to 11 are better organized since they present both the techniques and the experimental application in the IMIDs field. Thus, paragraphs 2 to 6 must be integrated with applicative examples of the described techniques.
Paragraph 4 should be included in the general part of the sequencing techniques description, or in a “consideration” paragraph, or even in the conclusion section.
Paragraph 12 should include an explanation on both machine learning and artificial intelligence: they must be considered as “techniques” and so they must be described a little more.
Another issue regards the numbering of the paragraphs: the consecutive numbers are not the best choice, it would really better to create sub-paragraphs. For example, paragraph 7 is the introduction of new-generation microscopy and the following paragraphs should be 7.1, 7.2, 7.3, and so on. In this way, the reader will understand better the organization and the subject of the work.
In general, I think that, right now, the manuscript is not fully addressing the aim exposed in the title and should be integrated to improve.
Author Response
We thank the referees and the editor for encouraging comments and suggestions. We are pleased that the editor and the reviewers found the work of interest. Please find below our considerations to the remaining concerns in a point-by-point manner.
Reviewer 1
The manuscript entitled “Getting closer to translational bioinformatics- The potential of OMICs technologies and artificial intelligence in the treatment of immune mediated inflammatory diseases” is a manuscript that tries to present, also from a technical point of view, a series of omics techniques and to explain their contribute in the field of immune mediated inflammatory diseases. The idea behind the work is good because it would be useful to have an overview of the most recent techniques and to understand how they could be effectively useful in the clinical application. However, I think that the manuscript results quite inhomogeneous and lacks a well-organized structure.
We thank reviewer 1 for the stimulating and supportive comments.
There are issues that should be addressed in order to make the work suitable for publication.
First of all, there is not a Conclusion/discussion paragraph: especially in a review, this part must be included because it must present the point of view of the authors on the subject of the review.
We reorganized the structure of the manuscript and included a discussion paragraph as suggested by the reviewer.
Another issue that arises is the difference in the matter between the “sequencing” paragraphs (from 2 to 6) and the “microscopy” paragraphs (from 7 to 11). Paragraphs 2 to 6 are just a description of the experimental approach presented: in some cases, the explanation is even too much detailed (for example from lane 219 to lane 227) and there are just a few examples of the various application of the techniques for the study of IMIDs. On the other hand, I think the paragraphs from 7 to 11 are better organized since they present both the techniques and the experimental application in the IMIDs field. Thus, paragraphs 2 to 6 must be integrated with applicative examples of the described techniques.
We added further examples for studies of IMIDs according to the reviewer´s suggestions. Detailed technical explanations like in lanes 219-227 have been deleted.
Paragraph 4 should be included in the general part of the sequencing techniques description, or in a “consideration” paragraph, or even in the conclusion section.
We re-organized paragraph 4 and included it partialy in paragraph 3 and in the conclusion section as suggested by the reviewer.
Paragraph 12 should include an explanation on both machine learning and artificial intelligence: they must be considered as “techniques” and so they must be described a little more.
We revised the paragraph according to the reviewer´s suggestions and integrated it in the conclusion section.
Another issue regards the numbering of the paragraphs: the consecutive numbers are not the best choice, it would really better to create sub-paragraphs. For example, paragraph 7 is the introduction of new-generation microscopy and the following paragraphs should be 7.1, 7.2, 7.3, and so on. In this way, the reader will understand better the organization and the subject of the work.
We revised the order of the paragraphs according to the reviewer´s suggestions.
In general, I think that, right now, the manuscript is not fully addressing the aim exposed in the title and should be integrated to improve.
We revised the title of the manuscript according to the reviewer´s suggestions.
Reviewer 2 Report
Theoretically, the manuscript is well-written, and the presentation is good. Unfortunately, the study rationale is unclear and the study aim too, which represents the main limitation. Indeed, it seems the aim "This review sheds light on the background of this time lag in the technological context, and discusses existing opportunities as to how the introduction of personalized medicine for the treatment of IMIDs can succeed" is too vague and conceptual as a purpose. The more concrete the study aims, the higher the scientific community and healthcare system will be influenced by.
Although the Authors have included several paragraphs that could be per se of some interest, the paper has no impact on clinical practice and does not report concrete insights in the diagnosis and treatment of immune-mediated diseases, as the title promised. It seems that the Authors did a brief summary of some technologies, but scientific publications cannot be only a summary of a certain topic; oppositely, they should allow drawing conclusions on a given topic as per the influence on clinical outcome and practice in the real world.
It could be sound to propose the manuscript to some technologies ad/or methodologies specialized Journal.
Author Response
We thank the referees and the editor for encouraging comments and suggestions. We are pleased that the editor and the reviewers found the work of interest. Please find below our considerations to the remaining concerns in a point-by-point manner.
Reviewer 2
Theoretically, the manuscript is well-written, and the presentation is good. Unfortunately, the study rationale is unclear and the study aim too, which represents the main limitation. Indeed, it seems the aim "This review sheds light on the background of this time lag in the technological context, and discusses existing opportunities as to how the introduction of personalized medicine for the treatment of IMIDs can succeed" is too vague and conceptual as a purpose. The more concrete the study aims, the higher the scientific community and healthcare system will be influenced by.
We thank reviewer 2 for the stimulating and supportive comments. We revised the description of the study aim according to the reviewer´s suggestion.
Although the Authors have included several paragraphs that could be per se of some interest, the paper has no impact on clinical practice and does not report concrete insights in the diagnosis and treatment of immune-mediated diseases, as the title promised. It seems that the Authors did a brief summary of some technologies, but scientific publications cannot be only a summary of a certain topic; oppositely, they should allow drawing conclusions on a given topic as per the influence on clinical outcome and practice in the real world. It could be sound to propose the manuscript to some technologies ad/or methodologies specialized Journal.
We included reports about concrete insights in the diagnosis and treatment of IMIDs according to the reviewer`s suggestion.
Reviewer 3 Report
A manuscript submitted by Anchang et al is a timely review discussing technological advances in the field of genomics. Authors have provided additional supporting evidences for the applications of these various techniques/methods including Artificial Intelligence (AI). Such comprehensive view of the field with applications is of great importance to the readers. Though this manuscript deserves to be published, authors are suggested to consider for following minor comments before publication:
- Authors are suggested to change the title of this review manuscript. After reading this review, the title doesn't much reflect the content. For example, authors explained OMICS technologies, but do not really described its potential. Do they propose anything new? Another example, Artificial Intelligence (AI) is the most catchy word nowadays but need to be "really" represented using the use-cases. Has it really done so?
- Section 3 - 5 are more technological explanations which is nice and can be easily read in the most of the reviews. Authors need to bring more substantial IMID perspective here. Otherwise it will be outdated.
- Figure: Authors should have one more figure to explain spatial technology or microscopy/imaging-based methods and their connection to systems biology through machine learning/AI.
- Figure 1 should have added the abbreviations.
Author Response
We thank the referees and the editor for encouraging comments and suggestions. We are pleased that the editor and the reviewers found the work of interest. Please find below our considerations to the remaining concerns in a point-by-point manner.
Reviewer 3
A manuscript submitted by Anchang et al is a timely review discussing technological advances in the field of genomics. Authors have provided additional supporting evidences for the applications of these various techniques/methods including Artificial Intelligence (AI). Such comprehensive view of the field with applications is of great importance to the readers. Though this manuscript deserves to be published, authors are suggested to consider for following minor comments before publication:
We thank reviewer 3 for the positive evaluation of the manuscript and the stimulating comments.
Authors are suggested to change the title of this review manuscript. After reading this review, the title doesn't much reflect the content. For example, authors explained OMICS technologies, but do not really described its potential. Do they propose anything new? Another example, Artificial Intelligence (AI) is the most catchy word nowadays but need to be "really" represented using the use-cases. Has it really done so?
We revised the title of the manuscript according to the reviewers suggestion.
Section 3 - 5 are more technological explanations which is nice and can be easily read in the most of the reviews. Authors need to bring more substantial IMID perspective here. Otherwise it will be outdated.
We included reports about concrete insights in the diagnosis and treatment of IMIDs according to the reviewer`s suggestion.
Figure: Authors should have one more figure to explain spatial technology or microscopy/imaging-based methods and their connection to systems biology through machine learning/AI.
We added an additional figure according to the reviewer´s suggestion.
Figure 1 should have added the abbreviations.
Abbreviations have been added in the revised version of the manuscript.
Round 2
Reviewer 1 Report
Thanks to the authors to have addressed the previous issues, ameliorating the manuscript. They give a clear overview of some of the omics techniques used. At this regard, since they rightly do not present all the omics techniques available, I think that they should comment this aspect at the
end of the introduction: they should remind to the readers that i.e. RNAseq, proteomics, metabolomics, WGS, WES are well-established techniques that gave a lot of results and that are actually the most used omics techniques, but also that they are continuously improving and evolving and that these improvements will be the matter of the manuscript. Then, I think that the
paper will be suitable for publication.

Author Response
We thank reviewer 1 again for the stimulating and supportive comments. We added one sentence at the end of the introduction as suggested.Reviewer 2 Report
Despite some efforts that could be appreciated, the Authors did not address the Reviewer’s comments. The manuscript is not suitable for publication in this Journal.
Author Response
No specific comments were made by Reviewer 2,
Editor